Behavioral, morphological, and ecological trait evolution in two clades of New World Sparrows (Aimophila and Peucaea, Passerellidae)

Cicero Carla ccicero@berkeley.edu 1
Mason Nicholas A. 1 4
Benedict Lauryn 2
Rising James D. 3
1 Museum of Vertebrate Zoology, University of California, Berkeley , Berkeley , CA , United States of America
2 School of Biological Sciences, University of Northern Colorado , Greeley , CO , United States of America
3 Department of Ecology and Evolutionary Biology, University of Toronto , Toronto , Ontario , Canada
4 Current affiliation:  Museum of Natural Science and Department of Biological Sciences, Louisiana State University , Baton Rouge , Louisiana , United States of America
Roper James
Electronic publication date: 2020 Jun 19
Publication date: 2020
Volume: 8
Electronic Location ID: e9249
Received 2019 Nov 22; Accepted 2020 May 7
Copyright: ©2020 Cicero et al.
Copyright year: 2020
Copyright holder: Cicero et al.
License: This is an open access article distributed under the terms of the Creative Commons Attribution License, which permits unrestricted use, distribution, reproduction and adaptation in any medium and for any purpose provided that it is properly attributed. For attribution, the original author(s), title, publication source (PeerJ) and either DOI or URL of the article must be cited.
License URL: https://creativecommons.org/licenses/by/4.0/

Keywords: Trait evolution, Behavior, Phenotype, Habitat, Phylogenetic signal, Avian duets, Passerellidae, Aimophila, Peucaea, Melozone

Funding: Museum of Vertebrate Zoology, University of California, Berkeley The molecular lab work for this study was supported by the Avian Genetics Fund of the Museum of Vertebrate Zoology, University of California, Berkeley. Financial support for this publication was provided by the Berkeley Research Impact Initiative (BRII) sponsored by the University of California Berkeley Library. The authors received no outside funding for this work. The funders had no role in study design, data collection and analysis, decision to publish, or preparation of the manuscript.

==============================
The New World sparrows (Passerellidae) are a large, diverse group of songbirds that vary in morphology, behavior, and ecology. Thus, they are excellent for studying trait evolution in a phylogenetic framework. We examined lability versus conservatism in morphological and behavioral traits in two related clades of sparrows (Aimophila, Peucaea), and assessed whether habitat has played an important role in trait evolution. We first inferred a multi-locus phylogeny which we used to reconstruct ancestral states, and then quantified phylogenetic signal among morphological and behavioral traits in these clades and in New World sparrows more broadly. Behavioral traits have a stronger phylogenetic signal than morphological traits. Specifically, vocal duets and song structure are the most highly conserved traits, and nesting behavior appears to be maintained within clades. Furthermore, we found a strong correlation between open habitat and unpatterned plumage, complex song, and ground nesting. However, even within lineages that share the same habitat type, species vary in nesting, plumage pattern, song complexity, and duetting. Our findings highlight trade-offs between behavior, morphology, and ecology in sparrow diversification.

Introduction

Behavioral, morphological, and ecological traits have been used historically to reconstruct evolutionary relationships, and many taxonomic groups were originally designated on the basis of shared, homologous characters (e.g.,  Hamilton, 1962; Storer, 1955; Wolf, 1977). For example, similarities in syringeal and cranial morphology, plumage, nesting behavior, and foraging mode were used to establish generic limits and hypothesize relationships in tyrannid flycatchers, one of the world’s largest and most diverse avian radiations (Lanyon, 1984; Lanyon, 1985; Lanyon, 1986; Lanyon, 1988a; Lanyon, 1988b). Likewise, Hamilton (1962) inferred species relationships and the origin of sympatry in the avian genus Vireo by comparing species-specific characteristics of distribution, habitat preference, foraging ecology, and external morphology. While contemporary studies of evolutionary relationships now rely largely on genetic data, studies of trait evolution in a phylogenetic framework continue to shed light on patterns of phenotypic evolution and diversification.

Trait lability, selective pressures (selective pressures), and random processes can cause traits to have different phylogenetic signals (Blomberg, Garland Jr & Ives, 2003; Grant & Grant, 2002). Furthermore, variation in the rate of phenotypic evolution, as well as multiple gains or losses of a trait, can lead to patterns of character variation that differ across taxa (Dodd, Silvertown & Chase, 1999; Mooers, Vamosi & Schluter, 1999). Morphology changes relatively slowly over time and is therefore thought to have stronger phylogenetic signal compared to behavior (Blomberg, Garland Jr & Ives, 2003). However, behavioral traits also can have phylogenetic signal, and variation among related taxa may result from conservative or labile evolutionary processes (De Queiroz & Wimberger, 1993; Cicero & Johnson, 1998; Cicero & Johnson, 2002a; Cicero & Johnson, 2002b; Price & Lanyon, 2004; Brumfield et al., 2007; Price, Friedman & Omland, 2007; Kamilar & Cooper, 2013; Barve & Mason, 2015; Anderson & Wiens, 2017; Fang, Mao-Ning & Chih-Ming, 2018).

Molecular phylogenies facilitate tests of how traits evolve within clades. For example, mitochondrial DNA (mtDNA) sequences for the Empidonax group of tyrant flycatchers are congruent with morphological, behavioral, and allozymic traits, although some behaviors such as nesting and migratory tendency have stronger phylogenetic signal than others such as foraging mode (Lanyon, 1986; Cicero & Johnson, 2002a). In Icterus orioles, song and plumage evolution are highly labile between species but conserved across the genus as a whole. Furthermore, Icterus songs are more labile than those in closely related oropendolas (Psarocolius, Ocyalus), which tend to have conserved song characteristics (Price, Friedman & Omland, 2007).

The New World sparrows (Passerellidae, formerly Emberizidae) are a large, diverse lineage of songbirds that are well suited for studies of trait evolution in a phylogenetic framework. Evolutionary studies of New World sparrows include analyses based on morphology, plumage, soft-part colors, behavior, egg coloration, allozymes, mitochondrial and nuclear gene sequences, and phylogenomic data from ultraconserved elements (Wolf, 1977; Patten & Fugate, 1998; Carson & Spicer, 2003; DaCosta et al., 2009; Klicka & Spellman, 2007; Klicka et al., 2014; Bryson Jr et al., 2016; Sandoval et al., 2017). Although comparisons among these studies are compounded by differences in taxon and character sampling, together they provide a valuable framework for studying sparrow evolution. Whether behavioral traits are more labile (Blomberg, Garland Jr & Ives, 2003) or conserved (Brumfield et al., 2007) than morphological traits in sparrows remains an open question that deserves study.

One especially interesting group of New World sparrows is the historical genus Aimophila, which has been plagued by taxonomic uncertainty due to extensive morphological variation. Members of this group were united originally by characteristics of the bill, wings, tail, and feet (Swainson, 1837; Baird, 1858), but other ornithologists have long thought that they represent species from distantly related lineages (Ridgway, 1901; Dickey & Van Rossem, 1938; Storer, 1955; Wolf, 1977). Within the past decade, molecular data (e.g.,  DaCosta et al., 2009; Klicka et al., 2014; Sandoval et al., 2017) have clarified relationships and demonstrated polyphyly of the “Aimophila” group. This spurred a taxonomic revision (Chesser et al., 2010; Remsen Jr et al., 2010) that placed species formerly classified as Aimophila into one of three genera (Aimophila, Peucaea, Rhynchospiza), recognized the alliance of “Aimophila” quinquestriata with the genus Amphispiza, and moved some taxa from the genus Pipilo to Melozone. Furthermore, species of Aimophila, Melozone, and Pipilo form a clade separate from Peucaea and Rhynchospiza.

Prior to molecular phylogenetic studies, species relationships in Aimophila sensu lato were hypothesized based on detailed study (Wolf, 1977) of behavioral, morphological, and ecological differences that grouped taxa into one of three ecological “complexes”: (1) Haeomophila complex (currently Peucaea: species ruficauda, sumichrhasti, humeralis, mystacalis, carpalis), which radiated in lowland thorn scrub forests of western Mexico and the Pacific lowlands of Central America, and are characterized by simple songs, chatter duets derived mostly from primary songs, prenuptial molt, raised nests, heavy bills, patterned adult plumages, juvenile plumages more similar to adults than in other groups, and mostly delayed skull ossification; (2) botterii complex (also Peucaea: species aestivalis, botterii, cassinii), which occupy weedy, open country in Central and North America, have dull plumages (often with yellow at the bend of the wing), are migratory with more pointed wings, sometimes have spotted first-year plumages, and sing complex songs with chitter duets; and (3) ruficeps complex (currently Aimophila: species ruficeps, rufescens, notosticta), which radiated in pine-oak woodlands of Mexico and Central America and have similar primary songs, chatter duets not derived from primary song, and similar plumage patterns with rusty head stripes or caps. Wolf (1977) also described the geographic distribution of each species, and noted that range sizes vary within ecological complexes. On the basis of behavioral similarities, Wolf (1977) suspected a close relationship between the ruficeps complex and towhees in the genus “Pipilo” (now Melozone)—a hypothesis since supported by molecular data (Carson & Spicer, 2003; DaCosta et al., 2009; Klicka et al., 2014). He was uncertain about the placement of another species, quinquestriata, because of its unique plumage and song traits, and did not include two South American species (currently Rhynchospiza: species stolzmani and strigiceps) in his study.

The detailed phenotypic analysis by Wolf (1977) provides an opportunity to revisit questions about trait evolution (Maddison, 1994) within the two clades of Aimophila (plus Melozone and Pipilo) and Peucaea (plus Rhynchospiza) in a modern phylogenetic comparative framework. In this study, we focus on whether behavior, morphology, and/or ecology exhibit phylogenetic signal in these two clades, and extend these ideas to a larger group of New World sparrows. Specifically, we ask which traits identified as informative by Wolf (1977) are phylogenetically conservative or labile. We also use these data to assess the extent to which behavioral and morphological traits are associated with open (grassland) versus closed (arid scrub or pine oak) habitat types. If we find a strong association, then species in the same habitat type (i.e., same ecological group) may have been subjected to either phylogenetic niche conservatism (Pyron et al., 2015) or similar selective pressures that drive local adaptation to the environment (Lenormand, 2012). In either case, we would predict that traits are more conserved among species in similar habitats than among those in different habitats.

Material & Methods

DNA sequencing

We constructed an independent phylogeny of sparrows with a focus on all species formerly in the genus Aimophila, which are now divided into three genera: Peucaea (22 samples), Aimophila (8), and Rhynchospiza (4). We also included samples of Pipilo (9) and Melozone (4) because of their relatedness to the ruficeps complex based on both behavioral (Wolf, 1977) and molecular (DaCosta et al., 2009) data. In addition, we sequenced two individuals of Amphispiza quinquestriata, which has been classified either in Aimophila (Dickinson, 2003) or Amphispiza (Chesser et al., 2010), along with Amphispiza bilineata (3) and a former congener Artemisiospiza belli (2) that is now placed in a monotypic genus (Klicka & Spellman, 2007; Chesser et al., 2010; Klicka & Banks, 2011). Using other phylogenetic studies of “Aimophila” and passerellid sparrows (Carson & Spicer, 2003; DaCosta et al., 2009; Klicka et al., 2014; Bryson Jr et al., 2016) as a framework for comparison, we added species in 10 other sparrow genera: Arremon (4), Chondestes (2), Oriturus (2), Spizella (4), Pooecetes (2), Ammodrammus (4), Ammospiza (2), Passerculus (2), Arremonops (2), and Zonotrichia (2). In total, we included 80 samples from 43 species in the Passerellidae and 4 non-sparrow outgroups (2 Parulidae, 2 Icteridae; Table S1).

We extracted genomic DNA from tissue using a modified salt extraction procedure (Miller, Dykes & Polesky, 1988), and PCR-amplified and sequenced four protein-coding mitochondrial genes (cyt-b, ND2, ATPase 8, COI) and three nuclear gene regions (intron 5 of transforming growth factor beta 2 [TGFb2] and beta-fibrinogen [Fib5], recombination activating gene RAG-1) using various combinations of primers (Table S2). We focused on the core ingroup taxa and putative allies for all loci (total of 5,344 bp: 3,495 mtDNA, 1,849 nDNA), and added mtDNA sequences from GenBank to fill out the taxon sampling (Table S1). PCR-amplification and sequencing were generally successful except for a few samples at some loci. We amplified DNA in 25 µL reactions with a mixture of 2 µL dNTPs (2 mM), 2.5 µl BSA (10 mM), 1.5 µL of each primer pair (10 mM), 2.5 µL of buffer (10×) pre-mixed with MgCl2, 0.1 µL of Taq polymerase, 1 µL of DNA, and double-distilled water. Amplification steps included an initial denaturation at 93 °C for 4 min followed by 30–35 cycles of denaturation (93 °C for 30 s), annealing (42−50 °C for 30 s), and extension (72 °C for 45 s), and a final extension at 72 °C for 5 min. Reactions had at least one negative and often a positive control, and we visualized PCR products on agarose gels stained with ethidium bromide. Following amplification, we cleaned the PCR products with Exonuclease I and Shrimp Alkaline Phosphatase (ExoSAP-IT, US Biochemical Corp.), and sequenced the purified products in both directions using Big Dye terminator chemistry v. 3.1 and an AB PRISM 3730 DNA Analyzer (Applied Biosystems). We checked and aligned all sequences using CodonCode Aligner v. 4.0.3 (CodonCode Corporation).

Phylogenetic analyses

We constructed phylogenetic trees using all 80 ingroup and 4 outgroup samples (Table S1) with mitochondrial and nuclear loci by performing Bayesian and maximum-likelihood concatenated analyses alongside species-tree inference. For the concatenated analyses, we first identified the best-performing model of sequence evolution for each locus and codon position for gene regions via Akaike Information Criterion with MrModeltest v. 2.3 (Nylander, 2004). We then constructed Maximum Likelihood (ML) phylogenies using RAxML v7.0.4 (Stamatakis, 2006; Stamatakis, Hoover & Rougemont, 2008), in which we performed 100 iterations of rapid bootstrapping while simultaneously finding the best tree in a single run with a GTR + I + G model of nucleotide substitution for each locus or gene region. We used BEAST v2.5.1 (Drummond & Rambaut, 2007; Drummond et al., 2012; Bouckaert et al., 2014) on the CIPRES Science Gateway (Miller, Pfeiffer & Schwartz, 2010) to conduct concatenated analyses in a Bayesian framework, in which we linked an uncalibrated clock model across loci but applied a separate HKY + I + G model of nucleotide substitution to each locus. We linked the tree prior for all loci and implemented a Yule model of speciation. We selected the Yule model because it is the simplest model of speciation, in which each lineage is assumed to have the same constant speciation rate, and is also appropriate for inferring phylogenies among species rather than among populations within species (https://www.beast2.org). We ran the BEAST analysis for 1 × 108 generations while sampling every 1,000 generations. We discarded the first 10% of sampled generations as burn-in, and assessed convergence and mixing by ensuring that ESS scores for each parameter exceeded 300 in Tracer v1.7.5.

We conducted a species-tree analysis using the *BEAST package within BEAST v2.5.1 (Bouckaert et al., 2014). For this analysis, we implemented a Yule speciation model and a constant population model with estimated population sizes for each gene tree and the resultant species tree. We ran the species-tree analysis for 1 × 109 generations and removed the first 10% as burn-in. For both the BEAST and *BEAST analyses, we subsequently generated maximum clade credibility trees from a thinned set of 5000 trees that was sampled every 20,000 or 200,000 generations, respectively.

Trait reconstructions

We scored 12 trait variables (9 binary and 3 multi-state) for each species (Table S3). Of these, 11 traits were described in detail by Wolf (1977) and we followed his scheme in assigning values as closely as possible. These included range size, typical habitat, plumage “brightness” (hereafter referred to as patterning), completeness of the postjuvenal molt, presence of a prenuptial molt, nest position, timing of skull ossification, group breeding, song structure, duetting, and duet type. We added geographic distribution as an additional trait in order to reconstruct its history in our focal clades. We assigned trait values based primarily on published information, which we took directly from Wolf (1977) for the species that he included, but we had to interpret and standardize definitions for some traits (e.g., range size, plumage patterning, song complexity) and for species not studied by Wolf. We used available audio recordings (Wolf, 1977 LP of audio recordings; Macaulay Library) to characterize song structure.

Binary traits used in trait reconstructions and tests of phylogenetic signal are described as follows:

(1) Plumage patterning: Unpatterned species are generally black or tan, but may show small patches of color or clearly delineated markings, such as the facial patterns on Aimophila sumichrasti. Patterned species have large patches of color that differ from the rest of the body.

(2) Postjuvenal molt: This molt is complete in species where individuals molt the entire plumage at this life stage, and incomplete in species where individuals molt only part of the plumage.

(3) Prenuptial molt (also known as prealternate molt, which occurs before breeding in certain birds): This molt is present in some species and absent in others.

(4) Skull ossification: Normal species have fully ossified skulls by the end of the first year. Skull timing is delayed in species where this process takes longer than one year.

(5) Nest position: Ground nesters typically build their nests on the ground. All species that build nests off the ground, regardless of height, are considered to have raised nests.

(6) Group breeding: Species where more than a pair of adults occur together during the breeding season are considered to have groups (Emlen, 1997). For example, Wolf (1977) characterized Aimophila ruficauda as having groups because he observed one female, one adult male, and additional first year males in the same breeding flock. This differs from other species where a single pair occurs on a territory. We scored group breeding as present only if the species frequently or regularly breeds in groups.

(7) Song structure: Song structure determinations followed Wolf (1977), other published reports (e.g., Rodewald, 2015), and examination of sound files (Table S3). Simple songs consist of one to four note types, although the notes may be repeated many times. They include songs with consistent syntax, including those that begin with a few introductory notes followed by a trill. Complex songs include a variable array of frequency-modulated note types and syntactical constructions.

(8) Duetting: Two individuals duet when they time their vocalizations to occur simultaneously or alternatively in a predictable manner.

(9) Geographic distribution: Northern Temperate species have breeding ranges in North America. Middle American species breed from Mexico through Panama.

Multi-state traits used in trait reconstructions are described as follows:

(10) Range size: We followed Wolf’s (1977) characterization of species as having small, medium, or large breeding ranges, which we measured from his published distribution maps as ca. 240 km long at the longest diameter, 240–800 km long, and over 800 km long, respectively. This trait, in combination with distribution, reflects geographic patterns of diversity and environmental tolerance (Stevens, 1989). Some species such as Aimophila notosticta, which is confined to the mountains of central and northern Oaxaca, have much more restricted ranges than other widespread taxa.

(11) Habitat: Arid scrub species coincide with Wolf’s (1977) “thorn scrub” category and live mostly in dry environments characterized by low, bushy vegetation. Pine-oak species live in woodlands that may be dominated by pine and/or oak trees. Grassland species live in open environments with predominantly grassy, herbaceous vegetation. Although this character has three states, we converted it to binary for trait correlation analyses. We designated grassland as “open” habitat, and both thorn scrub and pine-oak as “closed” habitat (Boncoraglio & Saino, 2007).

(12) Duet type: We used Wolf’s (1977) named duet types to indicate duet structure, and only included character states for species that he coded because these designations are somewhat subjective. Squeal duets have broadband elements that sound like squeals. Chitter and chatter duets have similar brief broadband ticking elements. Aimophila carpalis gives a unique “warbled” (Wolf, 1977) duet.

We used the BEAST maximum clade credibility tree (ML showed the same topology) with the concatenated mtDNA and nDNA dataset to estimate character transition rates and reconstruct ancestral character states. We reconstructed character states on our tree with all samples as well as in the two clades of Peucaea (colored blue in Fig. 1) and Aimophila plus the closely related genera Melozone and Pipilo (colored pink in Fig. 1). We performed ancestral state reconstructions of our categorical traits using a model-fitting approach that allowed for polymorphic character states within the package corHMM (Beaulieu, Oliver & O’Meara, 2017). Polymorphic character states were assigned likelihoods following the methods of Felsenstein (2004), with each possible character assigned an equal probability. This allowed us to estimate the phenotype of ancestral nodes while incorporating uncertainty in species’ phenotypes that were based on missing or incomplete data. We implemented an ‘equal rates’ model, in which transition rates between any character state were assumed to be equal with an upper bound of 100, while the character state of the root for each group was estimated following differential equations put forth by Maddison, Midford & Otto (2007) and FitzJohn, Maddison & Otto (2009). After estimating the transition rate matrix, we subsequently calculated the marginal likelihood states at each node.

Figure 1 Concatenated analysis of phylogenetic relationships in the Passerellidae.

Maximum clade credibility tree for concatenated analysis (4 mitochondrial and 3 nuclear genes) and all taxa using BEAST. Asterisks indicate posterior probability values of 0.95 or higher. Taxa in black were originally classified as Aimophila prior to recent revision (DaCosta et al., 2009). The two clades outlined by boxes are the focus of detailed analyses of trait evolution. Bird illustrations are provided courtesy of Lynx Edicions.

We used Pagel’s (1994) correlation method in Mesquite (Maddison & Maddison, 2003) to test Wolf’s hypothesis that individual traits vary in association with habitat for Peucaea. In this group, we tested for associations of prenuptial molt, nest location, song structure, and plumage patterning with open and closed habitat. We did not test other traits such as molt or skull ossification because we had no a priori predictions about their relationships, and we lacked the required information for all taxa. Because this test requires binary character states, we did not test non-binary traits. All members of the Aimophila clade live in closed habitat, so within-clade tests for effects of habitat are uninformative; however, we tested for a relationship between song complexity and plumage patterning in that group. We ran tests with 10 extra iterations over 10,000 simulations. Extra iterations implement additional searches within the maximum likelihood framework, and the simulation number is used to estimate statistical significance, with higher numbers above 100 returning better p-value estimates based on simulation output (Maddison & Maddison, 2018). Because the tests of Wolf’s specific hypotheses were done on small samples, we followed up on some of the associations they revealed by using the same correlation method to relate song with plumage and habitat use for all species in the tree. We were unable to evaluate additional traits in this way because of missing data across the full tree.

We examined trait lability among all species in the full tree for a subset of behavioral and morphological traits by calculating the D statistic, which is suitable for binary, categorical traits (Fritz & Purvis, 2010), using the function phylo.d within the caper package in R (Orme, 2018). Binary traits included in these analyses included plumage patterning, postjuvenal molt, prenuptial molt, skull ossification, nest position, group breeding, song structure, and duetting. The bounds of the D statistic depend on the number of tips in the phylogenetic comparative analysis, but in general, more negative values imply stronger phylogenetic signal (Fritz & Purvis, 2010). The D statistic is calculated by comparing the sum of observed sister-clade differences in the evolutionary history of the binary trait (∑dobs) to simulated data sets of sister-clade differences generated by randomly shuffling the tip values of the phylogeny (∑dr) and another simulated data set generated by Brownian motion (∑db). Thus, D is comparable across data sets such that when D is equal to 1, the binary trait in question has a phylogenetically random distribution across the tips of the phylogeny. In contrast, when D is equal to 0, the distribution of binary values across the tips is equal to that expected under Brownian motion (Fritz & Purvis, 2010). Furthermore, values of D can fall outside of the range of 0 to 1, such that negative values indicate phylogenetic conservatism beyond that expected by Brownian motion, while values greater than 1 indicate phylogenetic dispersion beyond that expected by random shuffling of tip values (Fritz & Purvis, 2010). This method also allows one to calculate two separate one-tailed probabilities (i.e., p values) that the observed D statistic is greater than 0 and less than 1. For each trait, we omitted taxa with unknown or ambiguous character states.

Results

Sequence variation

The complete data set of 84 individuals from 47 species and up to 5,344 bp of sequence contained 1,740 variable (32.6%) and 1,546 (28.9%) potentially parsimony-informative sites. The two clades for which we reconstructed character states had 1,324 (24.8%) variable and 1,188 (22.2%) parsimony-informative sites. Average nucleotide composition for the mitochondrial genes cyt-b and ND2 were similar to values reported in previous studies of this group and related taxa (Klicka & Spellman, 2007; DaCosta et al., 2009), with an excess of cytosine (36%) and a deficiency of guanine (10–13%). Average uncorrected sequence distances among core taxa for the mitochondrial gene regions were 11% in Peucaea (6.6–14.8%) and 4.9% in Aimophila (3.7%–6.1%). The mean distance between Aimophila and the closely related genera Melozone and Pipilo was 9.1% (range of 7.7% to 11.5%).

Phylogeny

Maximum likelihood (Fig. S1) and Bayesian methods (Fig. 1) of phylogenetic reconstruction produced similar phylogenetic hypotheses, with the strongest support obtained for the concatenated analysis of mtDNA and nuclear sequence data (Fig. 1). With the exception of three genes (ATPase 8, Fib 5, TGFb2), the best model was GTR + I + G for the data partitioned by loci, mtDNA partitioned by codon position, and combined mtDNA and nDNA sequences. With all samples combined, taxa grouped into two lineages that received high to moderate support in the phylogenetic analyses. The first lineage included Peucaea, Rhynchospiza, Arremonops, and Ammodramus. Within that lineage, the eight species of Peucaea formed a monophyletic group that was strongly supported and distinct from Rhynchospiza and the other genera. The second lineage included species in multiple genera, with a strongly supported clade that united species retained in Aimophila with species of Melozone and Pipilo. The species quinquestriata was sister to Amphispiza bilineata in a lineage that included Chondestes and Spizella, and those taxa were distant to the clade containing Aimophila. The species tree analyses generated a phylogeny that was concordant with the concatenated approaches and many of the same relationships were recovered (Fig. S2). However, the resultant species tree did not have strong posterior probability values for the large majority of nodes, which likely reflects the relatively small number of loci and the small number of individuals per species used in the coalescent-based species tree analysis (Camargo et al., 2012; Fig. S2). A species tree constructed with many more loci also was not able to resolve all relationships within the family (Bryson Jr et al., 2016).

Trait reconstructions Peucaea and Aimophila clades

Ancestral state reconstructions (Fig. 2 through Fig. 5) show that both the Peucaea and Aimophila clades originated in Middle America (Fig. 2), with some members of each clade shifting their ranges northward into the Northern Temperate zone. Aimophila species descended from a common ancestor that is predicted to have a large geographic range and a preference for pine-oak (closed) habitat (Figs. 2 and 3). We were unable to reconstruct the geographic range and habitat preference of ancestral Peucaea species unequivocally.

Figure 2 Trait reconstructions for geographic distribution and range size in two focal clades.

Maximum-likelihood based trait reconstructions of geographic distribution and range size among the Peucaea and Aimophila clades with Character states are indicated by different shades of gray, and the probability of each character state is indicated by the proportion of that shade on the nodes.

Figure 3 Trait reconstructions for habitat type and nest placement in two focal clades.

Maximum-likelihood based trait reconstructions of habitat type and nest placement among the Peucaea and Aimophila clades with Character states are indicated by different shades of gray, and the probability of each character state is indicated by the proportion of that shade on the nodes.

Molt patterns, plumage patterning, and timing of skull ossification showed different histories in the two clades. The ancestral species in both clades had partial postjuvenal molts, but they differed in the presence (Peucaea) or absence (Aimophila) of a prenuptial molt (Fig. 4). Prenuptial molt has been lost once in Peucaea, and gained twice within the broader Aimophila clade. Evolutionary patterns of plumage coloration likewise differed between clades (Fig. 4). The ancestral Peucaea had unpatterned plumage, and there has been a single transition to patterned coloration in one descendant lineage. In contrast, the Aimophila clade shows more uncertainty, with multiple probable transitions between unpatterned and patterned plumage. While the ancestral Aimophila species had normal skull ossification timing, the skull timing of the Peucaea ancestor is uncertain and there is diversity in this trait among modern lineages (Table S3). Three Peucaea species form a clade with normal skull timing, three species form a clade with delayed skull timing, and a third clade is split with one species in each category.

Figure 4 Trait reconstructions for plumage patterning and prenuptial molt in two focal clades.

Maximum-likelihood based trait reconstructions of plumage patterning and presence or absence of a prenuptial molt among the Peucaea and Aimophila clades with Character states are indicated by different shades of gray, and the probability of each character state is indicated by the proportion of that shade on the nodes.

Figure 5 Trait reconstructions for song structure and duet type in two focal clades.

Maximum-likelihood based trait reconstructions of song structure and duet type among the Peucaea and Aimophila clades with Character states are indicated by different shades of gray, and the probability of each character state is indicated by the proportion of that shade on the nodes.

Ancestral state reconstructions of behavioral traits also showed different patterns. Most species in the two clades live in pairs and do not form larger social groups (Table S3). The only exceptions are P. ruficauda and P. humeralis. Because their close relative P. mystacalis does not form groups, the presence of groups in P. ruficauda and P. humeralis may represent separate gains of the trait or a single gain with a subsequent loss of the trait in P. mystacalis. The ancestral nest type for Peucaea is a raised nest (Fig. 3), while the ancestral nest type for Aimophila is equivocal. However, members of both clades use both nest locations. Simple songs are the ancestral condition in both clades, with complex songs evolving once among the Peucaea group and twice among the Aimophila group (Fig. 5). Many members of both clades produce vocal duets (Fig. 5). Duetting clearly represents an ancestral condition among Peucaea species that is highly conserved, while duets have been lost at least twice within the Aimophila clade (A. notosticta, Pipilo). Furthermore, duet type shows phylogenetic conservatism in acoustic structure (Fig. 5). Peucaea species all sing rapidly modulated “chitter”, “chatter”, or “warble” duets, while all members of the Aimophila group with well-described duets produce broadband “squeal” duets.

Trait correlations

Pagel (1994)’s correlation tests showed that preference for closed habitat is correlated with patterned plumage (p = 0.011) and simple songs (p = 0.010) in the Peucaea clade. In contrast, open habitat preference is correlated with unpatterned plumage (p = 0.0069) and complex songs (p = 0.011), as well as with ground nesting (p = 0.010), in this clade. Open habitat use is not correlated with prenuptial molt (p = 0.11). All species in the Aimophila clade occur in closed habitat, where they exhibit a negative association between vocal and visual signals such that simple song is correlated with patterned plumage (p = 0.021).

Across our full tree (Fig. 1), unpatterned coloration is correlated with transitions into open habitats (p = 0.026), mirroring the results within our two focal clades. Song structure did not correlate with transitions to or from open (p = 0.746) habitat. Plumage patterning correlated with song complexity such that patterned birds tended to have simpler songs (p = 0.045) across all species in our tree.

Measures of trait lability

To examine the lability of behavioral and morphological traits among sparrows through time, we estimated character state changes for eight traits using our full tree that included a broader sampling of our two focal clades and related taxa without missing data (Table 1). We found a range of estimated D values, indicating variation in phylogenetic signal among behavioral and morphological traits. For the behavioral traits we examined, presence or absence of duetting behavior exhibited the strongest phylogenetic signal (D =  − 1.72), while group breeding behavior exhibited the weakest phylogenetic signal (D = 1.17). Among the morphological traits, skull ossification exhibited the strongest phylogenetic signal (D =  − 1.21), while plumage patterning exhibited the weakest phylogenetic signal (D = 0.56). On average, phylogenetic signal was stronger among the four behavioral traits (mean D =  − 0.71) compared to the four morphological traits (mean D =  − 0.31).

Table 1 Estimates of phylogenetic signal and the sum of sister-clade differences in binary behavioral and morphological traits using the phylogeny depicted in Figure 1.

The D statistic indicates the amount of phylogenetic signal present in the binary trait. When D = 0, the phylogenetic signal of a given trait is equal to Brownian motion. When D = 1, trait evolution is random with respect to phylogeny. Thus, more negative D values indicate stronger phylogenetic signal and fewer changes between sister clades, while higher D values indicate less signal and more changes between sister clades. Values in the PD>0 column indicate the probability that trait evolution exhibits less phylogenetic signal compared to a null distribution of values under Brownian motion. Values in the PD<1 column indicate the probability that trait evolution exhibits more phylogenetic signal compared to a null distribution of values when trait evolution is random with respect to phylogeny. Each null distribution was generated with 1,000 permutations.

	# of Taxa	Sum of sister-clade differences	D statistic	PD>0	PD<1	
Behavioral traits	
Group breeding	45	8.44	1.17	0.16	0.51	
Nest position	39	9.69	−1.05	0.93	0.00	
Song type	46	10.00	−1.25	0.96	0.00	
Duetting	39	7.28	−1.72	0.99	0.00	
		Mean = 8.85	Mean = −0.71			
Morphological traits	
Postjuvenal molt	35	6.83	−0.93	0.84	0.01	
Prenuptial molt	42	15.13	0.34	0.32	0.09	
Plumage brightness	47	27.54	0.56	0.00	0.00	
Skull ossification	33	5.69	−1.21	0.88	0.00	
		Mean = 13.80	Mean = −0.31			

Discussion

Phylogenetic relationships of the Peucaea and Aimophila clades

We found similarities and differences from prior phylogenies of New World Sparrows (DaCosta et al., 2009; Klicka et al., 2014; Bryson Jr et al., 2016; Sandoval et al., 2017). Overall, our results support division of the former “Aimophila” into Peucaea, Rhynchospiza, and Aimophila, but the phylogenetic details differ. For one, we found Peucaea carpalis and P. sumichrasti to be sister to the remaining Peucaea with over 95% posterior probability (PP) support in the concatenated analysis (Fig. 1), while DaCosta et al. (2009) could not resolve this relationship; however, support was lower in our species tree (Fig. S2) and in the maximum likelihood tree of Klicka et al. (2014). Another difference was in the clade containing Aimophila rufescens, A. ruficeps, and A. notosticta. While DaCosta et al. (2009) and Klicka et al. (2014) found strong support for a sister relationship between A. notosticta and A. ruficeps based on mtDNA when all three taxa were included, we recovered a sister relationship between A. ruficeps and A. rufescens using both mtDNA and nuclear markers in our concatenated analysis (PP >0.95). Our species tree analysis, on the other hand, was unable to resolve the relationships between these three taxa. The different studies all supported a sister relationship between Aimophila, Melozone, and Pipilo, although Sandoval et al. (2017) did not recover monophyly within Melozone (i.e., some species are more closely related to Aimophila than other congeners) with more intensive sampling of that genus. We also confirmed that quinquestriata is the sister to Amphispiza bilineata, although these taxa are separated by a deep branch and both are distantly related to both “Aimophila” and Artemisiopiza (formerly Amphispiza) belli; none of the prior studies included all three taxa in their analyses. Finally, we found Peucaea and relatives to be sister to other sparrows sampled, while Bryson Jr et al. (2016) found the Amphispiza lineage to be sister to other sparrows, including Peucaea, based on UCE sequence data. Together, these studies offer a compelling overview of species relationships among Aimophila, Peucaea, and related sparrow taxa, although additional work is needed to resolve some relationships. Furthermore, they clarify relationships in the three ecological complexes that Wolf (1977) defined, including support for a close affinity between the Aimophila ruficeps complex and species in the genus Melozone.

Trait evolution within the Aimophila and Peucaea clades

All species in the Aimophila and Peucaea clades have Middle American ancestors. The ancestor of the Aimophila clade had a large range size, but range size was equivocal in the Peucaea clade and reflected high variability among those species. Wolf (1977) noted that species in this clade had ranges centered around Mexico, with possible Middle America origins, and pointed out that closely related species varied in range size. Our analyses support these ideas and highlight the variability in range location and size within the group. Four of the eight Peucaea species have expanded (3) or moved (1) their ranges from ancestral Middle America to Northern Temperate locations. Six of the twelve Aimophila/Melozone/Pipilo species also have expanded (4) or moved (2) their ranges into Northern Temperate regions. Anecdotally, none of the species that showed range shifts are long-distance migrants, but northern temperate species tend to have larger ranges (Howell & Webb, 1995). These results fit with recent work showing that the common ancestor of all species in Passerellidae was likely a tropical endemic (Winger, Barker & Ree, 2014). The findings also support Rapaport’s rule, which states that high latitude species tend to have larger ranges than low-latitude species (Stevens, 1989; Cicero & Johnson, 2002b).

Ancestors of Aimophila and Peucaea sang simple songs and formed pair bonds. Subsequently, group living evolved only in Peucaea humeralis and P. ruficauda, while complex songs evolved three times and are now present in seven of our modern focal species (Fig. 5). Wolf (1977) used song and duet similarity as a justification for grouping species together, and our phylogeny supports those groupings while confirming that shifts in song form occur primarily between but not within groups. Likewise, Marshall (1964) concluded that voice is a good predictor of relationships within the “brown towhee” complex (Melozone fusca, M. crissalis, M. aberti, M. albicollis), especially when used with other attributes. Song structure is known to vary widely across avian species, and other work has shown that song traits may be both conserved and divergent within and among groups (Price & Lanyon, 2002; Price, Friedman & Omland, 2007; Snyder & Creanza, 2019). Importantly, Wolf’s (and hence our) divisions of songs into “simple” and “complex” reflect only two potential measures of complexity—syllable type diversity and syntax. Because we followed Wolf’s trait assignments, these categories are qualitative. More detailed and quantitative song-form analyses would be a valuable follow-up to this work, and might show that elements of song complexity are differentially conserved or labile through evolutionary time (Benedict & Najar, 2019).

Ancestral habitat use and nesting behavior varied between clades, as did skull ossification timing, molt patterns, and plumage. The Aimophila common ancestor might have had patterned plumage, while the Peucaea ancestor was likely unpatterned. Modern species in both groups show a range of plumage patterns, which appear to be relatively labile suggesting that color patterning can both appear and disappear. Similar trends have been found in other avian species and across birds more generally (Price, Friedman & Omland, 2007; Hofmann, Cronin & Omland, 2008; Dunn, Armenta & Whittingham, 2015; Maia, Rubenstein & Shawkey, 2016; Shultz & Burns, 2017; Marcondes & Brumfield, 2019), showing that evolution may favor elaborate plumage or drabness depending on selective pressures. In addition, there appears to be a negative association between plumage patterning and song complexity, both within our focal clades and across our full phylogeny. Two lineages that contain species with complex songs (Peucaea cassinii-P. aestivalis-P. botteri and Aimophila rufescens-A. ruficeps-A. notosticta) are characterized by unpatterned plumage, while species in other lineages with simple songs (e.g., Peucaea mystacalis, P. humeralis, P. ruficauda) have patterned plumage. Other studies on the evolution of plumage and song complexity in birds have shown that some groups (e.g., cardueline finches Badyaev, Hill & Weckworth, 2002) exhibit a similar trade-off whereas other groups (e.g., tanagers Mason, Shultz & Burns, 2014) do not show a correlation between song and plumage elaboration. Such mixed results suggest that the relationship between song and plumage likely depends on a variety of factors, which may include physiological processes (Shutler, 2010) or ecological interactions.

Song complexity may be greater in open versus densely vegetated habitats because of the acoustic properties of those habitats (Morton, 1975; Wolf, 1977; Wiley, 1991; Derryberry, 2009; Mason & Burns, 2015; Derryberry et al., 2018; Crouch & Mason-Gamer, 2019; but see Karin et al., 2018; Hill, Pawley & Ji, 2017). Within the Aimophila and Peucaea clades, we found that complex songs are significantly associated with open grassland habitat, and simple songs are associated with closed (arid scrub or pine-oak) habitat. Such a relationship may result from habitat structure, but might also arise because more grassland species (Peucaea botteri, P. cassinii, P. aestivalis) occur in Northern Temperate latitudes where they experience higher environmental variability, which is known to influence bird song complexity (Medina & Francis, 2012; but see Najar & Benedict, 2019). We did not, however, recover the same relationship when all species were included. Therefore, we have tentative support for Wolf’s (1977) hypothesis that habitat drives song features within the focal clades, but his observed trend is not universal. It is possible that the observed correlations between habitat and song within the Aimophila and Peucaea clades results from small samples sizes, because a small number of trait transitions drive these correlations (Maddison & FitzJohn, 2015).

Color evolution is often driven by habitat type, with natural selection favoring certain colors, patterns, or lack of patterning (Dunn, Armenta & Whittingham, 2015; Shultz & Burns, 2013; Marcondes & Brumfield, 2019; Miller et al., 2019). However, a global analysis showed that habitat does not predict plumage patterns across birds as a whole (Somveille, Marshall & Gluckman, 2016). We found that unpatterned plumage correlated with open grassland habitat among members of the Aimophila and Peucaea clades, as well as when trait correlation analyses were run using the full tree. Thus, unlike Wolf’s (1977) ideas about the influence of habitat on song, his hypotheses regarding habitat and plumage evolution appear to apply broadly within the Passerellidae. Unpatterned coloration can be advantageous for crypsis in open grassland habitats (Hill & McGraw, 2006). Our findings—along with studies of other specific groups such as woodpeckers (Miller et al., 2019) and ovenbirds (Marcondes & Brumfield, 2019)—suggest that the influence of habitat on plumage patterning may be clade-specific.

Lability versus stability of behavioral and morphological traits

Although behavioral traits are expected to be more labile than morphological traits (Blomberg, Garland Jr & Ives, 2003), we found that the behavioral traits identified by Wolf (1977) exhibited stronger phylogenetic signal across our full tree than the morphological traits (Revell, Harmon & Collar, 2008). In particular, prenuptial molt and plumage patterning showed low phylogenetic signal and high lability. This result is counterintuitive for prenuptial molt, because molt strategies in birds are integral to their life history (e.g., Terrill, 2017; Terrill, 2018) and are not predicted to be highly labile. In contrast, concordant with our findings, studies on diverse taxa have shown that plumage patterning is generally quite labile across avian clades (Omland & Lanyon, 2000). Lability in this trait is associated with a variety of biotic and abiotic attributes, such as variation in mating systems (Møller & Birkhead, 1994; Price & Whalen, 2009) and light environments (Shultz & Burns, 2013; Marcondes & Brumfield, 2019). The species studied here all have similar monogamous mating systems, but patterning was correlated with habitat across Passerellidae, providing a potential selective factor shaping patterning. Future work studying this variability would be informative.

Song structure, duetting, nest location, group breeding, skull ossification, and postjuvenal molt are all traits with strong phylogenetic signals. The most highly conserved trait was duetting, which was frequent across the tree but had few evolutionary origins. Both song structure and duet type tended to be conserved, such that close relatives used similar sounds. Complex song is often attributed to sexual selection (Andersson, 1994), while duetting is associated with pair-bond maintenance and territory defense (Logue & Hall, 2014). For song structure, the phylogenetic signal in our focal clades came primarily from the derivation and maintenance of complex song in two lineages (Fig. 5). Conservation of complex song is sometimes found in other groups (Price & Lanyon, 2002; Tietze et al., 2015; but see Price, Friedman & Omland, 2007). For this study, we followed Wolf (1977) in defining song complexity based on the number and variety of note types in the species-typical song. Although debate exists about what metrics of song best describe “complexity” (Pearse et al., 2018; Najar & Benedict, 2019; Benedict & Najar, 2019), increased complexity reflects higher syllable diversity in the species we studied and is conserved in related lineages. This result might suggest that closely related species are under similar selective pressures for maintenance of song structure, potentially relating to visual signaling or habitat as discussed above (Panhuis et al., 2001; Boncoraglio & Saino, 2007).

Duet vocalizations are derived and maintained in many of the focal species in our study. Avian duets have been shown to perform a range of functions, including joint resource defense, mate defense, and pair coordination (Hall, 2009; Dahlin & Benedict, 2014). Work on the genera Melozone and Peucaea has demonstrated that duets of different species have similar functions in resource defense, providing a possible selective pressure maintaining this trait (Benedict, 2010; Sandoval, Méndez & Mennill, 2013; Illes, 2015; Sandoval, Juárez & Villarreal, 2018). Similarly, studies of other New World avian clades have shown that vocal duet presence and form are often evolutionarily conserved (Mann et al., 2009; Mitchell et al., 2019). This pattern is likely driven by life-history traits such as monogamy, territoriality, and sedentariness, which are shown by many of the species included in our analysis (Benedict, 2008; Logue & Hall, 2014). Most strikingly, duet type (Fig. 5) in addition to duet presence is conserved, as noted by Wolf (1977). Our focal species therefore provide a valuable system for future analyses examining how territorial behavior throughout the year and the length of pair bonds might promote evolutionary stability in behavioral traits. Overall, the strong phylogenetic signal found for vocal traits and other behaviors, including nest location and group breeding, counters a general assumption that behavioral traits are more labile than morphological traits (Blomberg, Garland Jr & Ives, 2003).

Conclusions

Our study elucidated relationships among New World sparrows and showed that behavioral traits such as vocal duetting and nest placement can exhibit stronger phylogenetic signal than morphological traits. Habitat appears to be an important driver of trait evolution within Aimophila and Peucaea, but its influence is not consistent within the Passerellidae. While habitat does not predict song evolution reliably across New World sparrows, the correlations of unpatterned plumage with open habitats and complex songs does hold broadly in sparrows. Outcomes suggest that New World sparrows provide a fertile testing ground for future studies of avian trait evolution.

Supplemental Information

Figure S1 Maximum likelihood phylogeny inferred from concatenated data set using RAxML

Circles on the nodes correspond to bootstrap support values, in which white circles indicate nodes that received less than 50 bootstrap support, gray indicates nodes with between 50 and 70 bootstrap support, and black indicates nodes with strong support greater than 70 bootstrap support.

Click here for additional data file.

Figure S2 Coalescent-based species tree

Species inferred using *BEAST. Circles on the nodes correspond to posterior probabilities, in which white circles indicate nodes with less than 70 posterior probability, gray indicates nodes with between 70 and 95 posterior probability, and black indicates nodes with strong support greater than 95 posterior probability.

Click here for additional data file.

Table S1 Samples and Genbank numbers

List of species, specimens, and GenBank numbers used in this study. Institutional codes are: MMNH, Bell Museum of Natural History, University of Minnesota; CNAV, Colección Nacional de Aves, Instituto de Biología, Universidad Nacional Autónoma de México; KUMNH, Biodiversity Institute and Natural History Museum, University of Kansas; LSUMZ, Museum of Natural Science, Louisiana State University; MBM, Marjorie Barrick Museum, University of Nevada Las Vegas (specimens transferred to UWBM); MVZ, Museum of Vertebrate Zoology, University of California Berkeley; MZFC, Museo de Zoología de la Facultad de Ciencias, Universidad Nacional Autónoma de México; NCSM, North Carolina State Museum; ROM, Royal Ontario Museum; STRI, Smithsonian Tropical Research Institute; UWBM, Burke Museum of Natural History and Culture, University of Washington.

Click here for additional data file.

Table S2 Gene regions and primers

Primer names, sequences, and sources for 4 mitochondrial and 3 nuclear gene regions analyzed in this study.

Click here for additional data file.

Table S3 Species and character state data

List of species, character states, and sources for character state data used in this study.

Click here for additional data file.

Sakina Palida assisted with the molecular lab work. Rauri Bowie helped with analyses and comments, and Michael Patten and five anonymous reviewers provided comments that improved earlier drafts of this paper. The Macaulay Library provided recordings on a CD, and the University of California Berkeley Language Center digitized Wolf’s (1977) recordings from LP. We thank the curators and collections staff at the following museums for loans of tissue samples used in this study and for providing information on voucher numbers (Table S1): Biodiversity Institute and Natural History Museum, University of Kansas (Town Peterson, Mark Robbins); Burke Museum of Natural History and Culture, University of Washington (Sharon Birks, John Klicka); Colección Nacional de Aves, Instituto de Biología, Universidad Nacional Autónoma de México (Patricia Escalante); Field Museum of Natural History (John Bates, Shannon Hackett, David Willard); Museo de Zoología de la Facultad de Ciencias, Universidad Nacional Autónoma de México (Blanca Hernández, Adolfo Navarro); Museum of Natural Science, Louisiana State University (Donna Dittman, Fred Sheldon); and Royal Ontario Museum (Brad Millen).

Additional Information and Declarations

Competing Interests

Author Contributions

Animal Ethics

DNA Deposition

Data Availability

The authors declare that they have no employment with non-academic affiliations, and they do not have any competing interests.

Carla Cicero conceived and designed the experiments, performed the experiments, prepared figures and/or tables, authored or reviewed drafts of the paper, and approved the final draft.

Nicholas A. Mason and Lauryn Benedict conceived and designed the experiments, performed the experiments, analyzed the data, prepared figures and/or tables, authored or reviewed drafts of the paper, and approved the final draft.

James D. Rising conceived and designed the experiments, authored or reviewed drafts of the paper, contributed many of the tissue samples, and approved the final draft.

The following information was supplied relating to ethical approvals (i.e., approving body and any reference numbers):

IACUC approval was not necessary because we worked with archival specimens.

The following information was supplied regarding the deposition of DNA sequences:

All sequences are available in GenBank. The accession numbers, a list of the samples with their museum and catalog number are available in Table S1.

The following information was supplied regarding data availability:

The Nexus input files, maximum clade credibility tree generated with BEAST2, and R script written to perform comparative analyses are available at Dryad: Cicero, Carla; Mason, Nicholas; Benedict, Lauryn; Rising, James (2020), Behavioral, morphological, and ecological trait evolution in two clades of New World sparrows (Aimophila and Peucaea, Passerellidae), v6, UC Berkeley, Dataset, https://doi.org/10.6078/D16Q4W.

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
