# Peer review of "Behavioral, morphological, and ecological trait evolution in two clades of New World Sparrows (Aimophila and Peucaea, Passerellidae)"

_PeerJ, doi:10.7717/peerj.9249_

## Round 0.1 · original submission · Minor Revisions

The reviewers have overall liked your paper. They have identified multiple minor points for improvement. Also, it was pointed out that you have missed a great opportunity, having all the data at hand, to revise the taxonomy. Maybe this will give you some ideas for a future paper. Or, you may be able to add taxonomy to the current paper.

·

Basic reporting

The paper is written clearly, the tables and figures relevant and well done, and background information and context are sound.

Experimental design

The phylogenetic methods used are “industry standard” and thus ought not to draw any undue scrutiny.

Validity of the findings

To me, the results are what they are. In some instances the authors presented a priori expectations drawn from Wolf’s monograph. In other instances no a priori expectation was stated, yet results are of interest nonetheless. Some may quibble with coding of traits, but choices are transparent and, hence, the study repeatable.

Additional comments

Near the end of the Introduction, the authors remark that, with respect to traits and habitats, “If we find a strong association, then species in the same habitat type (i.e., same ecological group) may have been subjected to similar selective pressures from the local environment and shared trajectories of trait evolution within their lineage.” Another possibility is the process (not just pattern) of phylogenetic niche conservatism.

Results for the lability index confuse me. Probabilities shown in Table 1 are for D ≠ 1 and D ≠ 0, with certain values in boldface type that, if used per custom, are meant to draw attention to those considered to be statistically significant. But how can that be? As reported, ff the probability that D ≠ 1 is 0 for, say, nest position, then there is no chance D is not 1, which is another way to say D = 1. But that does not make sense, either, given that tabled values of D nearer to 1.0 have P higher (i.e., a bigger chance D ≠ 1) than values further from 1.0. And there is the matter of the plumage brightness result, which implies Schrödinger's D, in that there D = 1 and D = 0 at the same time. I take it, then, that the header really ought to be the probability D = 1 and D = 0? If that is true, then at least the plumage brightness result makes some sense: D definitely is neither 1 nor 0, which seems reasonable given the estimate is roughly midway between the two.

One broader question is whether the authors have any data on territoriality across the various species? They mention territoriality as a prospective predictor of duetting, which makes sense, but it could be particularly interesting to explore, perhaps in a future paper, whether tendency toward or type of duet is associated with pair bond duration and associated aspects of whether territories are defended by the male or by the pair.

Specific comments:

l. 104: It may reflect nothing more than personal preference, but when I see a genus or species name in quotes I assume it is meant ironically, to signal a dubious name. It would be clearer to say Aimophila sensu lato.

l. 240: breeds

l. 299: Pearson product-moment correlation?

l. 340: By definition, negative values are less than zero. Is something else meant here?

l. 470: None of the species in Aimophila (sensu lato) or Melozone or Pipilo is a long distance migrant, so this statement puzzles me.

l. 482: Not to nitpick, but the “Brown Towhee complex” is only part of Melozone and could be viewed as a single species pair (M. fusca and M. crissalis). As written, it is implied that the whole genus is in the complex.

ll. 493–494: Regarding “. . . reinforcing the notion that evolution does not always favor more elaborate plumage, but can also create drabness,” has there been a suggestion somewhere that elaborate plumage is a directed trait? (If so, holy crap.) I imagine virtually any clear-headed evolutionary biologist would respond “it depends”: strong sexual selection could favor elaborate plumage regardless of, say, increased parasitism risk, whereas strong natural selection for crypsis (which the authors mention later) in the face of strong predation pressure could favor drab plumage regardless or, say, female choice (or the choice could select to elaborate song instead, as perhaps hinted at by negative correlations reported herein).

Fig. 1: Why Ammodramus rather than Ammospiza for A. nelsoni and A. lecontei?

Reviewer 2 ·

Basic reporting

The authors of this manuscript analyzed the evolution of behavioral, morphological and ecological traits among the species of two clades within the New World Sparrows. They focused on the same species and traits considered by a paper from the 70s (Wolf 1977), which used such traits to infer phylogenetic relationships. Now, the authors use a molecular phylogeny to analyze evolution of the same traits in a more modern framework and adding some species.
Previous references and context provided are adequate. The general structure could be improved, as I found it hard to follow which species where considered for each analysis, and more importantly, inclusion/exclusions of species are not always properly justified.
The raw data shared could allow replication of the study, but how some of that data was produced is not completely clear (see comments below on how song was classified into "simple" or complex")

Experimental design

Some parts of the introduction still lead to the idea that one of the objectives of the paper is to use phenotypic/behavioral traits to infer phylogenetic relationships between the species, which is not what the authors did (see lines 47-49 and 128-129 in the Introduction).
I see the value in re-visiting classic studies, specially when they have produced a dataset still informative nowadays if re-analyze with modern statistical techniques (such as those that take into account phylogenetic relationships). However, some of the very interesting traits considered here, such as plumage pattern (characterized as present or absent) and song structure (divided into "simple" or "complex" in an obscure way), should be analyzed in a more rigorous and informative way than it was done 40 years ago. Adding a phylogeny is certainly important but not the only thing necessary to provide a “modern framework” when reconsidering a classic paper.
In the case of song, if authors decide to keep this oversimplified, binary classification for song structure, they should be more explicit about how they classified the songs of the species they had to analyze themselves. They say they were based on the methods by Wolf (1977), who states that “For this report the types of vocalizations will be divided primarily on the bases of their supposed function and internal complexity” but I could not find much information about how he did so, and therefore it is not clear how the authors of the present manuscript replicated this method for the recordings they analyzed themselves.

Validity of the findings

Previous reviewer raised concern about the validity of the conclusions considering that most analysis where based on a small subset of species. I do not think a “small” dataset in terms of number of species is a problem in itself, as long as traits are measured in a less simplistic way than it is usual for studies considering entire families or even orders. This is not case in this manuscript for some traits such as plumage pattern and song structure, which I suggest re-considering to better support the conclusions drawn from results obtained.
For example, analyzing song evolution by classifying it into either “simple” or complex” is an oversimplification. There is an ongoing debate of what makes a song more “complex”, so in its current way the relationship found between habitat and song structure seems uninformative. This level of simplification questions the validity of the conclusions made base on the obtained results. For example, many of the species in this family have trilled songs, and the separation between notes in a trill can be strongly influenced by the level of vegetation coverage in the habitat. Here, apparently all trilled songs have been classified as “simple”.

Additional comments

I can’t help thinking that this seems like a missed opportunity to build a more comprehensive phylogenetic hypothesis for the New World Sparrows. It is noticeable that none of the many previous studies on New World Sparrows systematics could provide the necessary framework for the 2 clades of interest and species in-between (due to differences in taxon and character sampling). A previous reviewer raised a similar point about including more species for the phenotypic data and I agree with the authors that “add more species” is usually an unfair a criticism that could potentially be done to any study. I also understand that the main objective of the authors was studying trait evolution and the molecular phylogeny was just a tool to achieve this, not an objective in itself. But building a more comprehensive phylogenetic hypothesis (after so many previous papers on the topic that still not providing the necessary framework for studying trait evolution in the entire family) certainly would have made the work stronger.

Reviewer 3 ·

Basic reporting

The article has been well structured after much editing, due to observations of other reviewers and from the editor of PeerJ, so one can see a better structure and clarity.

Experimental design

The experimental design is appropriate, both for phylogenetic reconstruction and for the analysis of the traits mapped and infered in the phylogeny in the two clades chosen.

Validity of the findings

I consider that a more robust phylogeny is offered here, after previous studies and a more complete sampling of taxa, so, this part represents an advance. The analyzes of the behavioral, ecological and morphological traits are satisfactory, being in the majority of the comparisons between the two clades, contrasting situations.

This paper is interesting for a more general audience, because it explores evolutionary issues that are relevant not only for this group of birds, but for others that have similar habitat, behavior or morphological characteristics.

Additional comments

I only noted a few small spelling mistakes in the following lines:

138 … traits remain an open question, however, we do not have a priori [without “and”]
464 Trait evolution within the Aimophila and Peucaea clades [a missing a in Peucaea]
607 Adolfo Navarro

External reviews were received for this submission. These reviews were used by the Editor when they made their decision, and can be downloaded below.

---

## Round 0.2 · Minor Revisions

As you may also be going through some tough times, I hope you understand that these days, impediments may occur and I hope you bear with us. Only one reviewer responded this second round and overall, I think the study has improved considerably. However, the writing style is both a bit verbose and not as clear as it could be.

Rather than list all the details here, I will simply enclose my annotated copy of the manuscript with the many comments I made. Every place I commented you will find a note that explains my position and my thoughts, except for the few places I simply deleted words without comment. Also note, I did include several comments on the content, the logic, the inclusion (or exclusion) of some details, and so these comments are not merely editorial.

I did not exhaustively comment on the text, but I tried to choose to comment on the patterns that you tend to have in your writing. Where I simply marked through words or phrases, those were redundant. Otherwise, I tried to explain how the writing style could be both clearer and briefer. I would like to request that as you read my comments, you consider the implications in your writing and try to find other places that you can improve the style. In all the examples I provide, I substantially reduced the number of words without losing any of the meaning. Brevity and clarity are the goals. I would suggest that you also find additional parts of your text that could be more brief and clear. Please feel free to edit your text based on these ideas and your own style.

In general, if possible, it is unnecessary to state something along the lines of "So and so (date) said this and such" when you might have just as easily said "This and such (So and so, Date)." (If you understand my shorthand.) The difference is my sentence emphasize the pattern or process while yours emphasizes the authors. Reading is more fluid and text is less wordy.

I hope you find all my comments useful and constructive.

·

Basic reporting

no comment

Experimental design

no comment

Validity of the findings

no comment

Additional comments

Title: Behavioral, morphological, and ecological trait evolution in two clades of New World Sparrows (Aimophila and Peucaea, Passerellidae)
Authors: Carla Cicero, Nicholas A. Mason, Lauryn Benedict, and James D. Rising

I appreciate the authors’ responses to the few comments I offered on my initial review of this manuscript. I have little new to add, and I continue to feel this paper is scientifically sound and will be of general interest. Its importance lies, I believe, in its basic goal to quantify and test ideas presented in Wolf’s seminal monograph from nearly a half-century ago. Wolf’s monograph is a showpiece in how to wed thinking in evolutionary ecology to phylogeny. It was ahead of its time.

I have but a single comment on this revision. The expanded explanation in the Methods of the D statistic is good, especially in light of Fritz and Purvis’ less than clear explication when they first presented the statistic. I nonetheless cannot wrap my head around results presented in Table 1. In the original paper, Fritz and Purvis presented similar results with associated p-values. In their Table 3 one can see the two null hypothesis tests (for D > 0 and D < 1; i.e., tests of Brownian and of random) and can make sense of associated p-values relative to the value of D itself. For instance, if D is small, then the first test may be “n.s.” whereas the second may be significant (i.e., the null hypothesis of D = 1 was rejected). Results in Table 4 are less readily interpreted, unless negative values of D are, in effect, spurious, by which I mean that values < 0 are treated statistically as = 0 when the null hypothesis test is conducted. I imagine this is a function of how the statistic was implemented in the R package, and as such perhaps no further explanation is needed. Even so, a glance at Table 4 can be jarring if p-values were interpreted by the reader as, say, strength of evidence (i.e., in a Bayesian way).


–Michael A. Patten
Oklahoma Biological Survey
University of Oklahoma

External reviews were received for this submission. These reviews were used by the Editor when they made their decision, and can be downloaded below.

---

## Round 0.3 · accepted · Accept

There are a few minor details that you need to check In the trait reconstruction figures, at least in my PDF, the names of the species sometimes run together and the letters overlap or the species is not separated from the genus, or both. Also, where you have written - Multiple colors on nodes indicate the probability of each character state - I think you could improve by something along the lines of "Character states are indicated by different shades of grey, and their probabilities as the proportion of that shade." Or, something along those lines.

External reviews were received for this submission. These reviews were used by the Editor when they made their decision, and can be downloaded below.